# Differences in Compositions of Oral and Fecal Microbiota between Patients with Obesity and Controls

**DOI:** 10.3390/medicina57070678

**Published:** 2021-06-30

**Authors:** Tomasz Stefura, Barbara Zapała, Tomasz Gosiewski, Oksana Skomarovska, Alicja Dudek, Michał Pędziwiatr, Piotr Major

**Affiliations:** 12nd Department of General Surgery, Faculty of Medicine, Jagiellonian University Medical College, 31-501 Cracow, Poland; tomasz.stefura@gmail.com (T.S.); skomarovska.o@gmail.com (O.S.); ala.ddudek@gmail.com (A.D.); michal.pedziwiatr@uj.edu.pl (M.P.); 2Department of Clinical Biochemistry, Faculty of Medicine, Jagiellonian University Medical College, 31-501 Cracow, Poland; barbara.zapala@uj.edu.pl; 3Division of Molecular Medical Microbiology, Department of Microbiology, Faculty of Medicine, Jagiellonian University Medical College, 33-332 Cracow, Poland; tomasz.gosiewski@uj.edu.pl; 4Centre for Research, Training and Innovation in Surgery (CERTAIN Surgery), 31-501 Cracow, Poland

**Keywords:** obesity, microbiota, body mass index, BMI, microbiome

## Abstract

*Background and Objectives:* The aim of this study was to compare the differences in compositions of oral and fecal bacterial microbiota between patients with morbid obesity and normal-weight controls. *Material and Methods:* This was a prospective cohort study. The study included group 1 (patients with BMI ≥ 40 kg/m^2^) and group 2 (patients with BMI from 18.5 to 24.9 kg/m^2^). Our endpoint was the analysis of the differences in compositions of oral and fecal microbiota between the groups. Oral swabs and fecal samples were collected from the patients. The analysis of microbiota was conducted using next-generation sequencing. *Results:* Overall, the study included 96 patients; 52 (54.2%) were included in group 1, 44 (39.8%)—in group 2. In group 1, oral microbiota included significantly more bacteria from genera *Veillonella*, *Oribacterium* and *Soonwooa*, whereas, in group 2, *Streptobacillus*, *Parvimonas* and *Rothia* were more common. Fecal microbiota in group 1 included more *Bacteroides*, *Odoribacter* and *Blautia* and group 2 was more abundant in *Ruminococcus*, *Christensenella* and *Faecalibacterium*. *Conclusions:* Both oral and fecal gastrointestinal microbiota differs significantly among patients with severe obesity and lean individuals.

## 1. Introduction

The prevalence of obesity has been steadily increasing in the past decades [1,2]. Risk factors of severe obesity include, for instance, age between 40 and 59 years, female sex and Hispanic race [3]. It should be remembered that excessive body weight is associated with the development of multiple comorbidities, including type 2 diabetes, cardiovascular diseases and several types of cancer [4]. Additionally, various diseases feature a more severe course among patients with obesity. This results in both lower quality of life and increased risk of premature death [5,6].

The microbiota of the digestive tract (microorganisms living in the digestive tract) has become a popular subject of research [7]. It is involved in inducing an inflammatory and metabolic response through multiple neural, immune and endocrine pathways [8,9]. The abnormal gastrointestinal microbiota may play an important role during the development of obesity by multiple mechanisms [10,11].

Changing the composition of the intestinal microbiota in individuals with obesity increases intestinal permeability and activates the immune system, which leads to chronic inflammation, which additionally increases the risk of obesity-related diseases, including type 2 diabetes [12].

Our hypothesis was that patients with BMI over 40 kg/m^2^ must have a significantly different microbiota from patients fitting in the normal BMI criteria since microbiota can contribute to the development of obesity. 

The aim of this study was to compare the differences in compositions of oral and fecal microbiota between patients with morbid obesity and normal-weight controls to identify bacteria potentially associated with morbid obesity.

## 2. Materials and Methods

### 2.1. Study Design

This is a prospective cohort study. It was conducted at one academic center and a private medical facility between November 2018 and November 2019. The patients were divided into two groups: group 1 (patients with BMI ≥ 40 kg/m^2^) and group 2 (patients with BMI from 18.5 to 24.9 kg/m^2^). Group 1 was recruited at a bariatric center at an academic hospital whereas group 2 was recruited at the abovementioned private facility designing personalized health programs focused on preserving health and achieving longevity. It included patients voluntarily reporting for an examination. The basic demographic and anthropometric characteristics of group 1 and group 2 are presented in Table 1. Additionally, we conducted a separate comparative analysis of the subjects with BMI ≥ 50 kg/m^2^ and BMI ≤ 50 kg/m^2^ included in group 1. The cutoff point of 50 kg/m^2^ was set due to the fact that this group of patients is more likely to develop complex obesity-related comorbidities and poses a greater challenge for potential bariatric treatment. Separate analysis of those patients seems to be advisable considering major differences in clinical management of this group [13,14]. 

The inclusion criteria were as follows: age between 18 and 65 years, informed consent to participate in the study and conformance with the eligibility criteria either for group 1 or group 2. The exclusion criteria were as follows: administration of antibiotics or probiotics within 30 days prior to gathering biological samples, tooth decay, inflammatory gastrointestinal diseases, thyroid diseases, history of cancer, diseases impairing the immune system. The study was designed and described according to all the STROBE checklist points for observational studies [15].

The authors created a database concerning patients. The database included anthropometric and clinical data: age, sex, body weight and BMI, maximal BMI, main comorbidities of obesity and smoking.

### 2.2. Analysis of Endpoints

The endpoint was to analyze the differences between compositions of oral and fecal microbiota between group 1 and group 2.

### 2.3. Collection of Oral Swabs and Fecal Samples

Prior to the gathering of the biological material, the patients had fasted for 12 h. Swab samples were collected by the authors in sterile conditions. Dental prostheses were removed before obtaining oral swabs. The patients were instructed on how to gather fecal samples to avoid contamination. The stool samples were frozen after collection. The biological samples were held at −80 °C. All of the sample processing was conducted in sterile conditions. The samples were frozen within 15 min from collection. The precise description of the procedure for collection of oral swabs and fecal samples used at our center was included in our previous publication [16].

### 2.4. Bacteria Identification

Decontamination was conducted using 70% alcohol and UV radiation. Blank controls were used during bacterial DNA extraction. Library preparation was conducted at a different laboratory. The researchers wore appropriate clothing at all stages of sample processing. DNA was isolated using commercially available QIAGEN kits. The V3 and V4 regions were amplified using the primers specified for the forward and reverse regions selected based on the evaluation by Klindworth et al. of the 16S rRNA gene [17]. An Illumina MiSeq platform was used to perform dual index sequencing. The precise description of DNA isolation, library preparation and sequencing used at our center was included in the previous publications [16].

### 2.5. Statistical Analysis

STATISTICA v13 (Tulsa, OK, USA) was used to perform a statistical analysis. We presented the normally distributed data as the means ± standard deviation, non-normally distributed data—as the medians and the first and third quartiles. The Shapiro–Wilk test was used to verify distribution of the studied variables. Quantitative data were analyzed with Student’s *t*-test and Mann–Whitney U test; *p*-values below 0.05 were considered statistically significant. The taxonomic classification of the 16S rRNA-targeted amplicon reads was performed using the Illumina 16S Metagenomics workflow. Alpha diversity was presented using the Rényi index. Beta diversity was presented using the principal coordinates analysis. Linear discriminant analysis (LDA) effect size (LEfSe) was used in metagenomic analyses including the microbiota analysis. It was accurately described in our previous research [16]. LEfSe identifies features that are statistically different between biological classes. Then, it conducts additional tests to assess whether these differences are consistent with the expected outcomes. LEfSe was used to identify statistically significant differences in the relative composition of microbiota between the samples collected from the patients included in group 1 and group 2.

## 3. Results

### 3.1. Demographic Characteristics

Overall, the study cohort included 96 patients. Group 1 included 52 (54.2%) people and group 2 included 44 (39.8%). The mean age was 41.7 ± 10.6 years; 58 (60.4%) participants were female. The median weight was 112.5 kg (65–137.8 kg). The median BMI was 39.5 kg/m^2^ (22.8–50.4 kg/m^2^). Additional demographic characteristics are presented in Table 1.

### 3.2. Next-Generation Sequencing

We analyzed 96 oral samples—52 of the patients with severe obesity and 44 of the healthy controls. The total read count for the oral samples was 4,956,418. The maximum count per oral sample was 294,676. The minimum count per oral sample was 5433. Overall, 99.97% of the reads that passed quality filtering were classified to species among the oral samples. We analyzed 76 fecal samples—32 of the patients with severe obesity and 44 of the healthy controls. The total read count for the fecal samples was 5,624,763. The maximum count per fecal sample was 110,589. The minimum count per fecal sample was 22,638. Overall, 99.9% of the reads that passed quality filtering were classified to species among the fecal samples. The comparisons of alpha biodiversity of the oral and fecal samples are presented in Appendix A Beta diversity is presented in Appendix A and D for the oral and fecal samples.

### 3.3. Compositions of Oral and Fecal Microbiota

In group 1, phyla Firmicutes, Proteobacteria, Actinobacteria, Bacteroidetes and Fusobacteria constituted 40%, 24%, 16%, 13% and 6% of the bacteria present in the oral cavity, whereas in the case of fecal microbiota, phyla Firmicutes, Bacteroidetes and Actinobacteria constituted 60%, 30% and 7% of the average composition in the patients in group 1. In group 2, the composition of oral microbiota included phyla Firmicutes, Proteobacteria, Actinobacteria, Bacteroidetes and Fusobacteria which constituted 29%, 28%, 25%, 14% and 4% on average. Fecal microbiota in the patients in group 2 included phyla Firmicutes, Bacteroidetes and Actinobacteria which constituted 67%, 22% and 9% on average.

### 3.4. Differences in Oral and Fecal Microbiota in Group 1 vs. Group 2

Among the patients with obesity, oral microbiota included significantly more bacteria from genera *Veillonella*, *Oribacterium*, *Soonwooa* and other genera presented in Figure 1, whereas bacteria from genera *Streptobacillus*, *Parvimonas*, *Rothia* and other genera presented in Figure 1 were significantly more common in oral microbiota among the patients included in the control group. Fecal microbiota among the patients included in group 1 included significantly more bacteria from genera *Bacteroides*, *Odoribacter*, *Blautia* and other genera presented in Figure 2. The patients included in the control group had fecal microbiota more abundant in bacteria from genera *Ruminococcus*, *Christensenella*, *Faecalibacterium* and other genera presented in Figure 2.

### 3.5. Differences in Oral and Fecal Microbiota in the Patients with BMI ≥ 50 kg/m^2^ vs. < 50 kg/m^2^


Among the patients with BMI above 50 kg/m^2^, oral microbiota included significantly more bacteria from genera *Leptotrichia*, *Abiotrophia*, *Oribacterium* and other genera presented in Figure 3, whereas bacteria from genera *Soonwooa*, *Veillonella*, *Alloprevotella* and other genera presented in Figure 3 were significantly more common in oral microbiota among the other patients with obesity. Fecal microbiota in the patients with BMI above 50 kg/m^2^ included significantly more bacteria from genera *Romboutsia*, *Lactobacillus*, *Flavonifractor* and other genera presented in Figure 4. The other patients with obesity had fecal microbiota more abundant in bacteria from genera *Butyricimonas*, *Intestinimonas*, *Murimonas* and other genera presented in Figure 4.

## 4. Discussion

This publication attempts to compare bacterial microbiota composition in the oral cavity as well as fecal microbiota between patients with severe obesity and individuals with the normal body mass. We conducted this study based on the need for a better understanding of challenging diseases such as obesity and comorbidities related to it [18]. We believe that modern science is only beginning to uncover the impact of how microbiota can influence pathology of certain ailments. To accomplish that, we used the Next-Generation Sequencing (NGS) of bacterial 16S RNA—a cutting-edge culture-independent method of analyzing the presence of bacterial organisms in a given environment [19].

Our study includes a comparison of microbiota at two levels of the gastrointestinal tract, which was rarely included in previous studies. Therefore, it adds new insight to the available data on the relationship between gastrointestinal microbiota and obesity. We also included a comparative analysis of microbiota of patients with superobesity (BMI of ≥ 50). Microbiota was reported to be involved in the development of obesity. Its role and composition seem not be sufficiently described in the available literature.

Microbiota of the oral cavity plays an important role in maintaining homeostasis. However, it is highly sensitive and easily falls out of balance due to weak immune system or systemic diseases, which can have consequences for local and systemic health [20]. Multiple factors can influence the composition of oral microbiota including infectious pathogens, use of antibiotics, diet, nutrition, lifestyle and socioeconomic factors [21]. *Streptococcus mutans*, *Porphyromonas gingivalis*, *Staphylococcus* sp. and *Lactobacillus* sp. have been reported to be the most commonly found in the human oral cavity [22]. Intestinal microbiota plays an important role in nutrient metabolism, xenobiotic and drug metabolism, maintenance of structural integrity of the mucosal barrier, immunomodulation and protection against pathogens [23]. The most common bacteria present in the large intestine are Proteobacteria, Firmicutes, Actinobacteria and Bacteroidetes [24].

In the course of our analysis of oral microbiota of the patients with severe obesity, it was identified to be significantly more abundant in bacteria from phylum Firmicutes. This is consistent with the previously published research. For instance, Tam et al. reported phylum Firmicutes to be an independent significantly discriminative feature with an abundance of over four orders of magnitude among patients with obesity [25]. Goodson et al. reported in their study that all Firmicutes except *Eubacterium* sp. and *Gemella morbillorum* had a significantly greater abundance in overweight individuals [26].

The patients included in this study in group 1 had intestinal microbiota more abundant in bacteria from phylum Bacteroidetes, including Bacteroides and Odoribacter. The systematic review of the gut microbiota profile by Castaner et al. revealed that obesity is associated with different profiles of gut microbiota, but there is a general lack of consistency in the results between the studies [27]. This is probably caused by several factors, including different methodologies used in the published studies. The authors highlight the need for future research on this subject to draw relevant conclusions on the role of microbial diversity in the development of obesity. The study by Gao et al. of gut microbiota revealed nine species showing a higher relative abundance of species *Bacteroides plebeius*, *Parasutterella excrementihominis*, *Parabacteroides distasonis*, *Bilophila wadsworthia*, *Clostridium symbiosum*, *Megamonas funiformis*, *Allisonella histaminiformans*, *Prevotella stercorea* and *Oxalobacter formigenes* in children with obesity in comparison with the control group [28]. LEfSe analysis conducted by Alpert et al. indicated that *Blautia* and *Anaerotruncus* were significantly more abundant in patients without obesity whereas bacteria from genus *Parasuterella* were significantly enriched in patients with obesity [29]. Kasai et al. reported a lower abundance of Bacteroidetes and a higher Firmicutes/Bacteroidetes ratio in subjects with obesity as compared with subjects without obesity. Additionally, the diversity of bacterial microbiota was greater among patients with obesity. Subjects with obesity had a larger abundance of species *Blautia hydrogenotorophica*, *Coprococcus catus*, *Eubacterium ventriosum*, *Ruminococcus bromii*, *Ruminococcus obeum* whereas patients without obesity—of *Bacteroides faecichinchillae*, *Bacteroides thetaiotaomicron*, *Blautia wexlerae*, *Clostridium bolteae*, *Flavonifractor plautii* [30].

This study has several limitations. Firstly, the patients with obesity were enrolled to the study group at only one bariatric center and the control group was recruited at one private facility devoted to providing prophylactic care and maintaining good health. Therefore, generalization of our results should be done cautiously. The sample size of our study group was limited by the size of funding granted for this research. The difference between the results of this study and previous publications may be partially attributed to the sample size and recruitment conducted at one bariatric center. A larger study cohort would yield more precise results. Secondly, our results are, partially, outcomes of a comparative analysis, and the observed results may be associated with sequencing efficiency. However, the samples included in our study had a relatively large, comparable number of sequences, thus minimizing this source of bias. There was a high rate of smoking patients in group 1 (11.5%) vs. group 2 (0%), which could potentially affect the composition of oral microbiota. It is important to mention that all the patients were advised to stop smoking at least 3 months prior to swab collection. However, we did not have the possibility to verify the compliance.

Further studies should be conducted on a larger study cohort recruited at multiple centers. Moreover, the role of oral and fecal microbiota in the development of obesity needs investigation. Identifying the role of bacterial microbiota in the development and treatment of obesity could allow for designing a more comprehensive approach to its treatment.

## 5. Conclusions

The presented findings confirm that gastrointestinal microbiota differs significantly among patients with severe obesity and lean individuals in the case of fecal microbiota as well as of oral microbiota. In the case of oral microbiota, the group with obesity had a larger abundance of phylum Firmicutes. Fecal microbiota was significantly more abundant in Bacteroidetes. The normal-weight participants had oral microbiota enriched, inter alia, in phyla Proteobacteria and Actinobacteria, whereas fecal microbiota, in comparison, was more highly abundant in phylum Firmicutes.

## Figures and Tables

**Figure 1 medicina-57-00678-f001:**
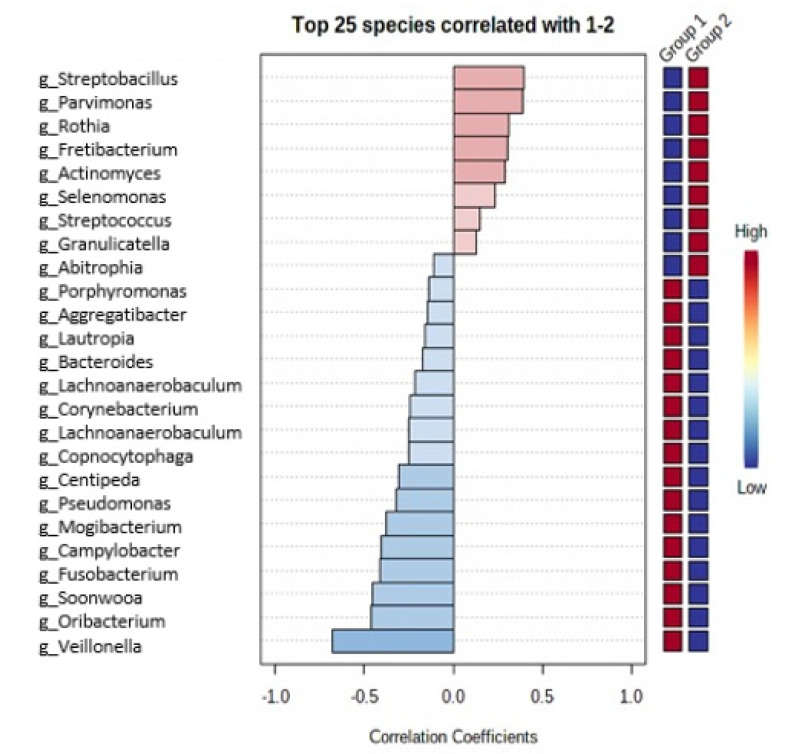
Differences in oral microbiota between group 1 vs. group 2.

**Figure 2 medicina-57-00678-f002:**
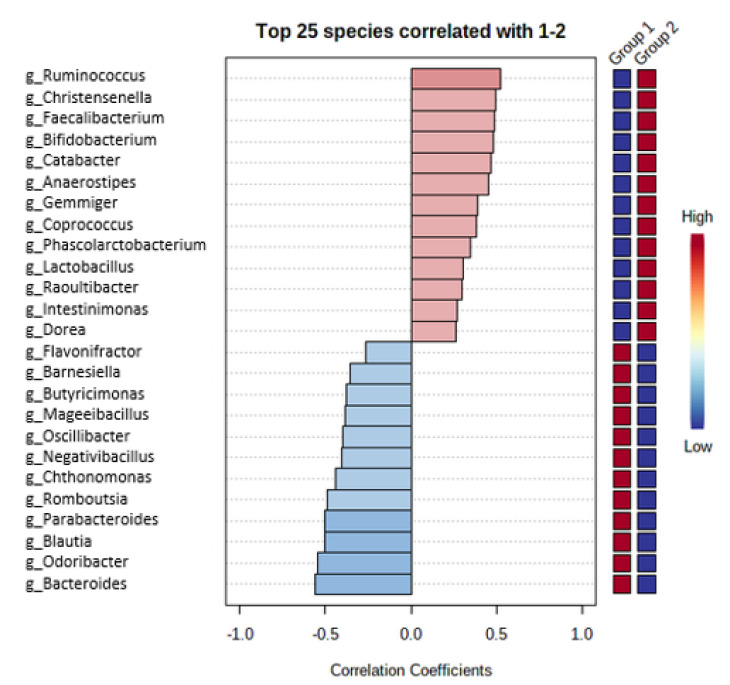
Differences in fecal microbiota between group 1 vs. group 2.

**Figure 3 medicina-57-00678-f003:**
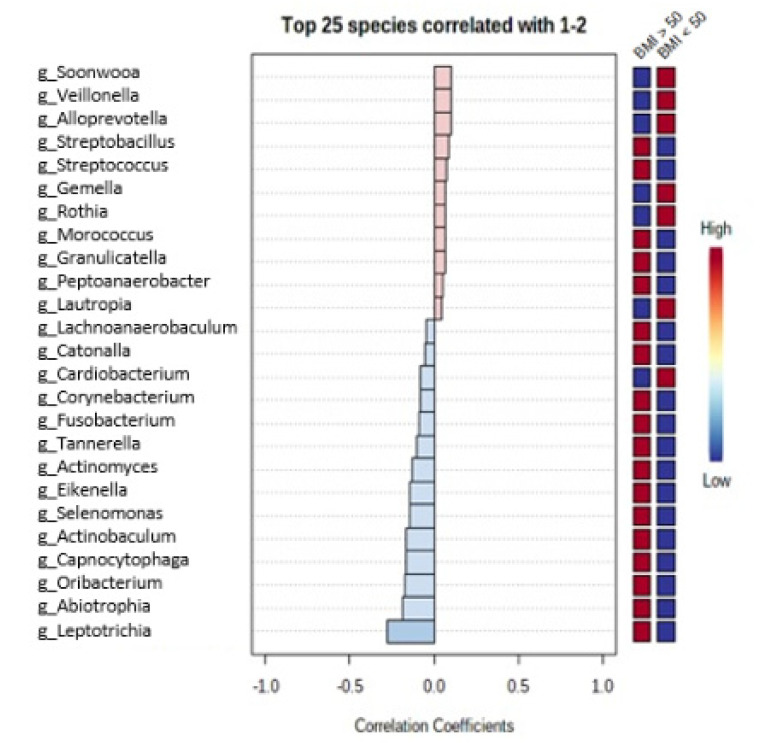
Differences in oral microbiota between the patients with BMI above 50 kg/m^2^ vs. the other patients with obesity.

**Figure 4 medicina-57-00678-f004:**
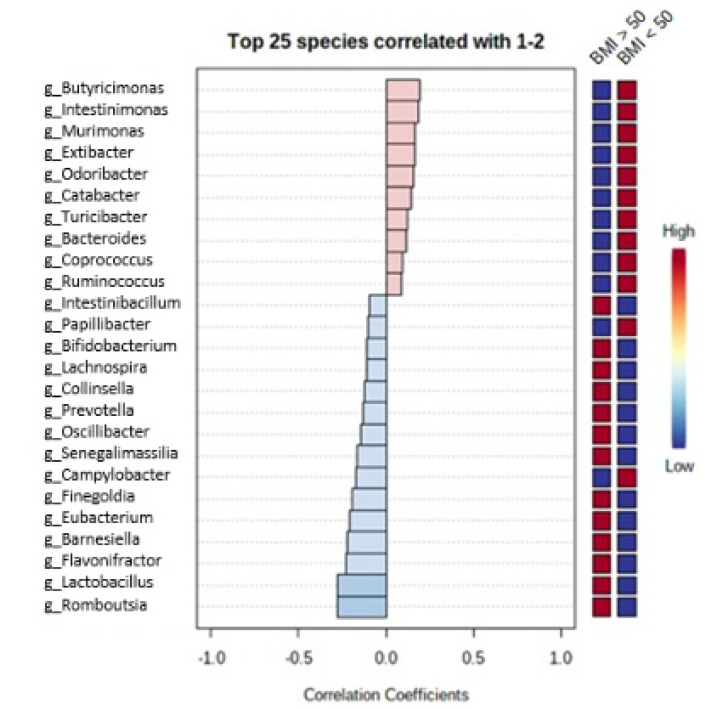
Differences in fecal microbiota between the patients with BMI above 50 kg/m^2^ vs. the other patients with obesity.

**Table 1 medicina-57-00678-t001:** Demographic characteristics.

Parameter	Total	Group 1	Group 2
Total, *n* (%)	96 (100)	52 (54.2)	44 (39.8)
Mean age, years ± SD	41.7 ± 10.6	43.3 ± 10.9	39.8 ± 10
Sex (female), *n* (%)	58 (60.4)	31 (59.6)	27 (61.4)
Median weight, kg (IQR)	112.5 (65–137.8)	136.5 (124–156.5)	61 (56.5–72.5)
Median BMI, kg/m^2^ (IQR)	39.5 (22.8–50.4)	50 (44.7–53.3)	22.2 (20.9–25.2)
Diabetes, *n* (%)	11 (11.5)	11 (21.2)	0
Diabetes complications, *n* (%)	3 (3.1)	3 (5.8)	0
Hyperlipidemia, *n* (%)	13 (13.5)	11 (21.2)	2 (4.5)
Steatohepatitis, *n* (%)	10 (10.4)	10 (19.2)	0
Hypertension, *n* (%)	37 (38.5)	35 (67.3)	2 (4.5)
Cardiovascular disorders, *n* (%)	9 (9.4)	9 (17.3)	0
Respiratory disorders, *n* (%)	9 (9.4)	9 (17.3)	0
Varicose veins, *n* (%)	9 (9.4)	9 (17.3)	0
Smoking, *n* (%)	6 (6.2)	6(11.5)	0

## Data Availability

The data are available from the authors upon reasonable request.

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
