# Peer review of "Differences in Compositions of Oral and Fecal Microbiota between Patients with Obesity and Controls"

_medicina, 2021, doi:10.3390/medicina57070678_

Round 1

Reviewer 1 Report

The authors studied to compare the differences in composition of bacterial microbiota in oral cavity and feces between patients with obesity and normal weight controls. As the result, the both of the oral and fecal microbiota differed among patients with obesity and controls. Although the manuscript is well-written, it appears little new findings. 

Major points:
1.    Since Turnbaugh et al reported gut microbiota as an additional contributing factor to the pathophysiology of obesity on the Nature in 2006, there have been a lot of reports regarding relation between human obesity and fecal or oral bacterial microbiota. The difference points of microbiota analyses in this study from the previous ones could have been attributable to the small enrolled patients size from the one bariatric center. 
2.    The smoking rate was significantly different between the Group 1 and 2. Could the the difference have been affected result of the microbiota analysis in the oral cavity ?

Minor points:
1.    The authors mention that they compared the differences in composition of bacterial microbiota in oral cavity and large intestine (Line-55 and -191). However, they did not take samples from large intestine but only feces. Thus, the description is not accurate. 
2.    The resolution of the Figures is quite poor. 
3.    Line-132; “NGS” should be spelled-out. And, if the authors use the abbreviation of NGS at the line-132, it is not need to spell-out at the line-195-6. 

Author Response

Dear Reviewer #1,

Thank you for reviewing our manuscript. Please find response to your comments below and in manuscript

.

Reviewer #1

The authors studied to compare the differences in composition of bacterial microbiota in oral cavity and feces between patients with obesity and normal weight controls. As the result, the both of the oral and fecal microbiota differed among patients with obesity and controls. Although the manuscript is well-written, it appears little new findings. 

Major points:
1.    Since Turnbaugh et al reported gut microbiota as an additional contributing factor to the pathophysiology of obesity on the Nature in 2006, there have been a lot of reports regarding relation between human obesity and fecal or oral bacterial microbiota. The difference points of microbiota analyses in this study from the previous ones could have been attributable to the small, enrolled patients’ size from the one bariatric center. 

Thank you for your comment. We have included that information in the limitations section of our manuscript.

  1.    The smoking rate was significantly different between the Group 1 and 2. Could the the difference have been affected result of the microbiota analysis in the oral cavity?

Thank you for your comment. There was a high rate of smoking patients in Group 1 11.5% vs 0% in Group 2, which could potentially affect the composition of the oral microbiota. It is important to mention, that all patients were advised to stop smoking at least 3 months prior to the swab collection. However, we did not have a possibility to verify the compliance. We have included that information in the limitations section of our manuscript.

Minor points:
1.    The authors mention that they compared the differences in composition of bacterial microbiota in oral cavity and large intestine (Line-55 and -191). However, they did not take samples from large intestine but only feces. Thus, the description is not accurate. 

Thank you for your suggestion. We have corrected our manuscript in accordance with your suggestion.

  1.    The resolution of the Figures is quite poor. 

Thank you for your suggestion. However, in accordance with suggestion of the Editor we have removed Figures with poor resolution.

  1.    Line-132; “NGS” should be spelled-out. And, if the authors use the abbreviation of NGS at the line-132, it is not need to spell-out at the line-195-6. 

Thank you for your suggestion. We have corrected our manuscript in accordance with your suggestion.

Reviewer 2 Report

Dear Editor,

Thank you for the opportunity to review the manuscript entitled: “Differences in compositions of the oral and the intestinal microbiota between patients with obesity and controls”. Authors are to be commended for putting together an interesting manuscript.

 Below you may find my comments:

2.1. Study Design

Please provide more detail regarding the controls subjects used in this study. In most similar studies it is difficult to define “normal controls”; thus, more detail regarding their background should be provided

  1. 3 Collection of fecal and swab samples

Please provide more detail regarding the process of collection samples. This is critical for the soundness of the study and should be clear for the readers of the journal

3.1. Demographic characteristics

Please remove all “p-values” from Table 1; the study was not designed to address this so these differences may be subject to chance.

Figures 1-4 should be presented in a higher resolution, they are not clear enough

Author Response

Dear Reviewer #2,

Thank you for reviewing our manuscript. Please find response to your comments below and in manuscript.

Reviewer #2

Dear Editor,

Thank you for the opportunity to review the manuscript entitled: “Differences in compositions of the oral and the intestinal microbiota between patients with obesity and controls”. Authors are to be commended for putting together an interesting manuscript.

 Below you may find my comments:

2.1. Study Design

Please provide more detail regarding the controls subjects used in this study. In most similar studies it is difficult to define “normal controls”; thus, more detail regarding their background should be provided

Thank you for your suggestion. We have corrected our manuscript in accordance with your suggestion and included a description of control group in the “Study Design” section of our manuscript.

2.3 Collection of fecal and swab samples

Please provide more detail regarding the process of collection samples. This is critical for the soundness of the study and should be clear for the readers of the journal

Thank you for your suggestion. We have included a reference to our previous publication describing in detail the procedure of collection of fecal and swab samples. To avoid self-plagiarism, we decided not to included this in presented article.

3.1. Demographic characteristics

Please remove all “p-values” from Table 1; the study was not designed to address this so these differences may be subject to chance.

Thank you for your suggestion. We have corrected our manuscript in accordance with your suggestion.

Figures 1-4 should be presented in a higher resolution, they are not clear enough.

Thank you for your suggestion. However, in accordance with suggestion of the Editor we have removed Figures 1-4.

Round 2

Reviewer 1 Report

The manuscript has been improved and is in a nice condition now.